# TWO-PHASE HEAD-SPECIFIC LoRA:
# BALANCING GLOBAL AND LOCAL ADAPTATION IN MULTI-HEAD ATTENTION

## ABSTRACT

Low-Rank Adaptation (LoRA) has become a standard technique for parameter-efficient fine-tuning of large pretrained models. However, applying a single low-rank update to the entire weight matrix assumes that all attention heads require the same adaptation, overlooking their diverse functional roles. Simply increasing rank under this setting often leads to diminishing returns and redundant parameter usage. To address this, we propose **Two-Phase Head-Specific LoRA (HS-LoRA)**. In the first phase, a global adapter—instantiated by any method that applies a shared update to the full multi-head weight matrix—absorbs broad domain-shift information common across heads. In the second phase, lightweight head-specific adapters refine residual variations, recovering individuality suppressed by the global update. This two-phase design disentangles adaptation into a shared global subspace and multiple head-specific residual subspaces, balancing efficiency with expressiveness. On the VTAB-1k benchmark, HS-LoRA yields substantial gains in Structured tasks (up to +7.59 pp) and shows complementary improvements when combined with global methods such as PiSSA and CaRA.

## 1 INTRODUCTION

Low-Rank Adaptation (LoRA) has emerged as a fundamental technique for parameter-efficient fine-tuning of large pretrained models, including large language models (LLMs) and vision transformers. Initially proposed by Hu et al. (2022), LoRA freezes the pre-trained weights and introduces trainable low-rank matrices into selected weight paths, allowing efficient adaptation with a fraction of the original parameter count, often less than 0.01% of the total parameters. Its practical advantages, such as negligible inference overhead and excellent downstream performance, have led to widespread adoption across diverse tasks in natural language processing and computer vision.

Numerous approaches have extended the original LoRA framework to improve performance and flexibility. For instance, QLoRA Dettmers et al. (2023) enables LoRA-based fine-tuning on 4-bit quantized LLMs, achieving state-of-the-art results on single-GPU setups while maintaining full precision accuracy. Meanwhile, AdaLoRA Zhang et al. (2023a) addresses the problem of uniform rank assignment in standard LoRA by dynamically assigning the rank budget based on the importance of each weight matrix. Moreover, PiSSA (Principal Singular values and Singular Vectors Adaptation) Meng et al. (2025) improves convergence by initializing LoRA adapters using the top singular vectors of the original weight matrix $W$ and freezing the residual components, thus focusing training on the most significant directions. Similarly, CaRA (Canonical Rank Adaptation) Lokvi et al. (2025) tensorizes ViT's multi-head attention and feed-forward layers and applies low-rank updates using Canonical Polyadic Decomposition (CPD) over the tensorised weight structure, enabling parameter-efficient adaptation across multiple dimensions.

Despite these advances, applying LoRA effectively often requires a deep understanding of the model's internal architecture. In practice, some models—especially certain implementations of Vision Transformers (ViTs)—use a *fused QKV structure*, in which a single weight matrix simultaneously generates the query, key, and value projections. For example, in PyTorch's `timm` library Wightman (2019), ViT models often represent QKV with a single matrix of shape $[3d, d]$, whereas Google's ViT implementation separates them as distinct matrices Dosovitskiy et al. (2021). If LoRA

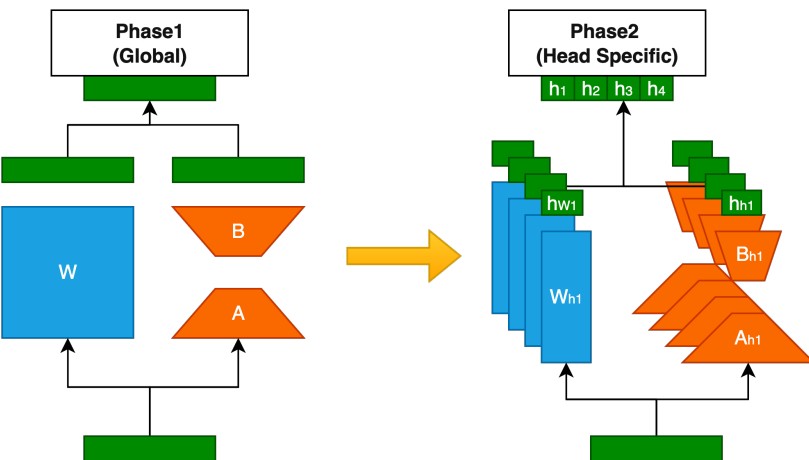

Figure 1: Illustration of our proposed **Two-Phase HS-LoRA**. In **Phase 1 (Global)**, a parameter-efficient fine-tuning method is applied uniformly across the entire projection matrix $W \in \mathbb{R}^{d \times d}$. This step is not restricted to standard LoRA; any adapter-based or LoRA-style method that produces a *shared update across all heads* (e.g., LoRA, QLoRA, AdaLoRA) can serve as the global update mechanism. Phase 1 primarily captures domain-shift information and broad adaptation signals that are common across heads. In **Phase 2 (Head-Specific)**, the globally updated matrix is further refined by attaching lightweight low-rank adapters to each submatrix $W^{(i)} \in \mathbb{R}^{d \times d_h}$ corresponding to an attention head. These per-head updates recover residual individuality and encode fine-grained variations suppressed by the global phase. Together, the two phases disentangle adaptation into a *shared global subspace* and multiple *head-specific residual subspaces*, thereby balancing efficiency with expressiveness.

is applied naively to a fused QKV matrix, the same low-rank update is shared across all three projections, potentially leading to degraded performance due to interference across subspaces. Indeed, in early experiments with such QKV-fused models, we observed lower performance when applying LoRA compared to models with separately managed Q, K, and V matrices. This suggests that LoRA's effectiveness is sensitive to how weights are structured internally and raises broader questions about its limitations when applied to concatenated or shared projection weights.

This insight led us to consider a more general problem: *How does LoRA behave when applied to a weight matrix that structurally encodes multiple independent sub-functions?* A compelling example of this occurs in **multi-head attention**, where the projection weights are logically divided across attention heads, each functioning semi-independently. In current LoRA implementations, a single low-rank update is shared across all attention heads. While this has shown empirical success, it implicitly assumes that the same transformation is appropriate for all heads—an assumption that may limit the granularity and effectiveness of adaptation.

To address this, we propose **Head-Specific LoRA**, a novel architecture that applies independent low-rank updates to each attention head. Concretely, given an attention projection matrix $W \in \mathbb{R}^{d \times d}$, we decompose it into $h$ per-head submatrices and associate each with a separate LoRA adapter. This design allows for *fine-grained, per-head adaptation*, capturing head-specific changes that would be averaged out in conventional LoRA. As we show through extensive experiments, Head-Specific LoRA consistently outperforms baseline LoRA across multiple tasks, providing a more expressive yet still efficient way to fine-tune multi-head attention models.

**However, our experiments also revealed key limitations.** While Head-Specific LoRA intuitively respects the individuality of each head, it did not always surpass existing LoRA-style approaches in performance and sometimes required more parameters. A closer inspection suggested that, although heads exhibit partially independent behaviors, many adaptation signals are in fact shared globally across heads. As a result, assigning fully independent adapters led to redundancy, with dominant shared components repeatedly captured in each head, leaving little capacity for truly head-specific

refinements. This observation led us to hypothesize that effective adaptation should capture both the *shared, globally relevant subspace* and the *residual, head-specific subspaces*. To this end, we propose a **two-phase strategy**: in the first phase, any existing LoRA-like updates are used to absorb broad, redundant domain shift information, and in the second phase, our method introduces orthogonal refinements that selectively encode head-specific individuality, thereby yielding more expressive and effective adaptation.

## 2 RELATED WORKS

### 2.1 PARAMETER-EFFICIENT FINE-TUNING

Parameter-efficient fine-tuning (PEFT) methods have emerged as essential techniques for adapting large pre-trained models to downstream tasks while minimizing computational overhead. Low-Rank Adaptation (LoRA) Hu et al. (2022) represents the foundational approach, introducing trainable low-rank matrices into pre-trained models while keeping original weights frozen. LoRA decomposes weight updates into the product of two smaller matrices with significantly reduced rank, enabling efficient adaptation with typically less than 1% of the original parameters. Several variants have extended LoRA's capabilities: QLoRA Dettmers et al. (2023) combines LoRA with 4-bit quantization for memory-efficient training, AdaLoRA Zhang et al. (2023a) dynamically allocates rank budgets based on weight importance, and DyLoRA Valipour et al. (2023) introduces adaptive rank scheduling. Other PEFT approaches include Adapter modules Houlsby et al. (2019) that insert bottleneck layers between transformer blocks, and Visual Prompt Tuning (VPT) Jia et al. (2022) that learns visual prompts for Vision Transformers. However, most existing methods apply uniform adaptations across entire weight matrices, potentially overlooking the internal structure and functional diversity within these matrices, which motivates our structured adaptation approach.

### 2.2 MULTI-HEAD ATTENTION

Multi-Head Attention (MHA) is a core component of Transformer architectures that enables models to attend to different representation subspaces simultaneously Vaswani et al. (2017). In MHA, the input is projected into multiple sets of query, key, and value matrices through separate linear transformations, with each attention head computing independent attention patterns before concatenation. Empirical studies have revealed that different attention heads naturally specialize in capturing distinct features and relationships, suggesting inherent functional diversity within the attention mechanism Voita et al. (2019); Clark et al. (2019).

In Vision Transformers, different heads naturally focus on various spatial and semantic aspects, with studies showing that heads specialize in different visual features and exhibit depth-dependent attention patterns Caron et al. (2021); Raghu et al. (2021b). Despite this natural diversity, current fine-tuning approaches, including standard LoRA, apply uniform updates across all attention heads, potentially limiting the model's ability to capture head-specific adaptations and failing to leverage the inherent structural organization of multi-head attention.

### 2.3 STRUCTURED AND MODULAR ADAPTATION

The limitations of uniform adaptation methods have motivated research into structured and modular fine-tuning approaches that align with the architectural properties of neural networks. Early work in modular adaptation includes task-specific adapter modules Rebuffi et al. (2017) and mixture-of-experts architectures Shazeer et al. (2017) that enable selective activation of different network components. These approaches recognize that different parts of a network may require distinct adaptations based on their functional roles.

## 3 PRELIMINARY

**Low-Rank Adaptation (LoRA)** LoRA is a parameter-efficient fine-tuning method that introduces trainable low-rank matrices into pre-trained models while keeping the original weights frozen Hu et al. (2022). Specifically, a pre-trained weight matrix $W_0 \in \mathbb{R}^{d \times d}$ is augmented with a low-rank decomposition $\Delta W = BA$, where $A \in \mathbb{R}^{r \times d}$ and $B \in \mathbb{R}^{d \times r}$, with $r \ll d$. During training, only $A$

and $B$ are updated while $W_0$ remains unchanged. The final output is computed as $h = W_0 x + BAx$, allowing the model to adapt efficiently without modifying its full parameter set.

Several variants have been proposed to improve or extend LoRA. QLoRA Dettmers et al. (2023) enables LoRA training on 4-bit quantized models with minimal performance loss. AdaLoRA Zhang et al. (2023a) dynamically allocates rank budgets across layers based on their importance. Dy-LoRA Zhang et al. (2023b) introduces a learnable rank schedule during training. These improvements aim to enhance LoRA's expressiveness, memory efficiency, and adaptability in low-resource settings.

**Multi-Head Attention in Transformers and Vision Transformers** Transformer-based architectures rely heavily on the multi-head attention (MHA) mechanism Vaswani et al. (2017). Given an input hidden state $x \in \mathbb{R}^d$, the module projects it into query, key, and value vectors using linear transformations: $Q = W_Q x$, $K = W_K x$, and $V = W_V x$, where $W_Q, W_K, W_V \in \mathbb{R}^{d \times d}$. These projection matrices are partitioned into $h$ heads:

$$W_Q = [W_Q^{(1)}; \ldots; W_Q^{(h)}], \quad W_Q^{(i)} \in \mathbb{R}^{d_h \times d}, \quad d_h = d/h,$$

and similarly for $W_K$ and $W_V$. Each head computes its own attention independently:

$$\text{Attention}^{(i)}(x) = \text{softmax}\left(\frac{Q^{(i)} K^{(i)^\top}}{\sqrt{d_h}}\right) V^{(i)},$$

and the outputs are concatenated and projected by $W_O \in \mathbb{R}^{d \times d}$:

$$\text{MHA}(x) = W_O \cdot \text{Concat}[\text{head}_1, \ldots, \text{head}_h].$$

This architectural design enables each head to attend to different subspaces of the input representation, improving the expressiveness of the model. In Vision Transformers (ViTs), prior work has shown that some heads specialize in local patterns (e.g., textures or object parts) while others focus on global relations Caron et al. (2021); Yamamoto et al. (2025); Raghu et al. (2021a); Lepori et al. (2024). However, empirical studies have also found significant redundancy among heads: many heads learn overlapping functions and can be pruned with minimal performance degradation Li et al. (2023); Yun & Ro (2024). This redundancy raises questions about whether a single low-rank update across all heads (as in standard LoRA) is sufficient to capture head-specific adaptations.

Furthermore, popular ViT implementations often fuse $Q$, $K$, and $V$ into a single weight matrix $W_{QKV} \in \mathbb{R}^{3d \times d}$ Wightman (2019). Applying LoRA naively to this fused matrix forces the same low-rank update to be shared across all three projections, introducing further entanglement that may harm performance. These observations motivate our method: a partitioned form of LoRA that can perform independent updates over head-level or other meaningful partitions of the weight matrix.

**Limitation of Standard LoRA.** In standard LoRA, a shared low-rank decomposition is added to the full matrix $W_Q$, resulting in:

$$W_Q^{\text{LoRA}} = W_Q + B_Q A_Q, \quad A_Q \in \mathbb{R}^{r \times d}, \quad B_Q \in \mathbb{R}^{d \times r}.$$

This update spans across all heads simultaneously, applying the same low-rank perturbation regardless of each head's functional specificity. That is, no head-specific nuance is intentionally captured.

This monolithic update is particularly problematic when:

- Attention heads specialize in different semantic or spatial roles;
- Some model uses a fused QKV matrix, where a shared LoRA update introduces cross-projection interference;
- The LoRA update interacts non-trivially with the ordering and semantics of head blocks.

**Adapter Merging.** An important practical property of many LoRA-family methods (e.g., LoRA, PiSSA) is that the trained low-rank adapters can be merged into the original weights after fine-tuning, eliminating the need for additional modules at inference time. Our HS-LoRA design preserves this property: despite applying per-head refinements, the resulting updates can also be consolidated into the base weights. This ensures that HS-LoRA does not introduce extra inference-time overhead beyond standard LoRA-style approaches.

# 4 OUR METHOD

## 4.1 OVERVIEW

We propose **Two-Phase Head-Specific LoRA (HS-LoRA)**, a framework that balances global efficiency with per-head expressiveness. Conventional LoRA applies a single low-rank update to the entire projection matrix, efficiently adapting shared representations but ignoring the functional diversity of individual heads. Head-Specific LoRA improves expressiveness by assigning independent adapters to each head, yet this often leads to redundant updates and increased parameter costs.

Our method integrates the strengths of both approaches by introducing a two-phase process. In **Phase 1**, we apply a global low-rank update that captures domain-wide and redundant components shared across all heads. This step is not restricted to LoRA itself: any parameter-efficient fine-tuning approach—such as LoRA, QLoRA, or AdaLoRA—can serve as the Phase 1 update as long as it produces a shared direction across heads. In **Phase 2**, we introduce lightweight head-specific refinements that restore individuality suppressed by the global update, ensuring that residual, head-specific variations are preserved.

By disentangling adaptation into a shared global subspace and multiple head-specific residual subspaces, Two-Phase HS-LoRA provides both efficiency and expressiveness. The following sections formalize the problem setup, present the two-phase architecture with its mathematical formulation, and describe implementation details for efficient training.

## 4.2 PROBLEM SETUP

We consider a multi-head attention projection where an input representation $x \in \mathbb{R}^d$ is transformed by a weight matrix $W \in \mathbb{R}^{d \times d}$. The matrix is partitioned into $h$ submatrices corresponding to attention heads:

$$W = \text{Concat}[W^{(1)}, W^{(2)}, \ldots, W^{(h)}], \quad W^{(i)} \in \mathbb{R}^{d \times d_h}, \ d_h = d/h.$$

Each head processes its own subspace, yielding

$$\text{head}_i = \text{softmax}\left( \frac{(W_Q^{(i)})^\top x \cdot (W_K^{(i)})^\top x}{\sqrt{d_h}} \right) (W_V^{(i)})^\top x.$$

Standard LoRA introduces a shared low-rank update across the entire matrix,

$$W^{\text{LoRA}} = W + BA,$$

where $A \in \mathbb{R}^{r \times d}, B \in \mathbb{R}^{d \times r}$. This design is parameter-efficient but forces all heads to share the same update, ignoring their functional diversity. Head-Specific LoRA addresses this by assigning independent updates per head:

$$W^{(i),\text{HS-LoRA}} = W^{(i)} + B^{(i)} A^{(i)}, \quad A^{(i)} \in \mathbb{R}^{r' \times d}, \ B^{(i)} \in \mathbb{R}^{d_h \times r'}.$$

Although more expressive, this approach increases parameter count and often redundantly captures globally shared signals.

## 4.3 TWO-PHASE HS-LoRA

To reconcile global and head-specific adaptation, we propose a **two-phase strategy**. The key idea is to first absorb broad, redundant information shared across heads using a global update, and then refine head-specific individuality with independent adapters.

**Phase 1: Global Update.** We introduce a low-rank adapter applied uniformly across all heads:

$$W^{\text{Phase1}} = W + BA,$$

where $A \in \mathbb{R}^{r \times d}, B \in \mathbb{R}^{d \times r}$. This captures domain shift and other signals all the heads share.

| Phase 1 method | w/o Phase 2 (HS-LoRA) | + Phase 2 (HS-LoRA) |
|---|---|---|
| LoRA | 77.14 | 89.39 |
| AdaLoRA | 82.10 | 89.90 |
| DoRA | 86.39 | 89.73 |
| PiSSA | 88.12 | **90.39** |

Table 1: Comparison of different Phase 1 PEFT methods when combined with Phase 2 HS-LoRA. Backbone: ViT-B/16, dataset: VTAB-1k SVHN. Phase 1 rank = 16, Phase 2 per-head rank = 4.

**Phase 2: Head-Specific Refinement.**   On top of the global update, we apply head-wise residual adapters:
$$W^{\text{Phase2}} = W^{\text{Phase1}} + \text{Concat}[B^{(1)}A^{(1)}, \ldots, B^{(h)}A^{(h)}],$$
where $A^{(i)} \in \mathbb{R}^{r' \times d}$ and $B^{(i)} \in \mathbb{R}^{d_h \times r'}$. These updates refine each head individually, enabling localized adaptation without duplicating globally relevant components.

**Final Formulation.**   The complete two-phase update is thus:
$$W^{\text{Updated}} = W + BA + \text{Concat}[B^{(1)}A^{(1)}, \ldots, B^{(h)}A^{(h)}].$$

## 4.4 TRAINING DETAILS

The parameter cost grows linearly with the number of heads in Phase 2, but this is offset by the ability to reduce the rank $r'$ relative to the global rank $r$. This allows a balanced allocation: high-rank global adapters for shared structure, and lightweight per-head adapters for residual refinements. Efficient implementation is achieved by storing adapter parameters in three-dimensional tensors,
$$A \in \mathbb{R}^{h \times r' \times d}, \quad B \in \mathbb{R}^{h \times d_h \times r'},$$
and computing the residual update via batched `einsum` contractions. This avoids Python-level loops and preserves the efficiency of standard LoRA.

## 5 EXPERIMENTS

### 5.1 OVERVIEW

We conduct experiments to evaluate the effectiveness of our proposed **Two-Phase Head-Specific LoRA (HS-LoRA)**. The experiments are designed to answer the following questions: (1) Among the LoRA family methods available in the PEFT library, which variant serves as the most effective Phase 1 when combined with our Phase 2 (HS-LoRA)? (2) Does the proposed Phase 2 (HS-LoRA) provide consistent improvements when stacked on top of strong Phase 1 baselines?

### 5.2 PHASE 1 CANDIDATES FROM THE LoRA FAMILY

To investigate the compatibility of our framework with different PEFT methods, we instantiate Phase 1 with several representative LoRA-family techniques from the PEFT library: LoRA Hu et al. (2022), AdaLoRA Zhang et al. (2023a), DoRA Liu et al. (2024), and PiSSA Meng et al. (2025). Phase 2 is fixed to our proposed HS-LoRA.

We conduct experiments on the SVHN dataset from VTAB-1k. In Phase 1, the rank is set to $r = 16$, while in Phase 2, each head receives a lightweight adapter of rank $r' = 4$. Table 5.2 reports the results. Across all Phase 1 methods, adding Phase 2 HS-LoRA consistently improves performance. LoRA gains +12.25 points, AdaLoRA +7.80, DoRA +3.34, and PiSSA +2.27, showing that HS-LoRA provides complementary benefits regardless of the baseline. The relative ranking is also preserved: stronger Phase 1 methods (e.g., PiSSA at 88.12) remain stronger after Phase 2 (90.39). We therefore adopt PiSSA as our Phase 1 choice in the following experiments, as its SVD-based initialization of global components aligns naturally with our head-specific refinement in Phase 2.

## 5.3 VTAB-1k Benchmarks

We evaluate Two-Phase HS-LoRA on the full VTAB-1k suite of 19 datasets. CaRA Veeramacheneni et al. (2025) publicly released pretrained checkpoints trained with their tensorised PEFT approach, which we adopt as Phase 1 baselines. In contrast, PiSSA Meng et al. (2025) did not release VTAB-1k models, so we trained Phase 1 PiSSA models ourselves with rank $r = 16$.

**Parameterisation.** For PiSSA + HS-LoRA, we apply Phase 2 adapters only to the **query and value** projections, since key projections contribute minimally to performance in prior studies. This reduces trainable parameters compared to full QKV injection. In CaRA, however, the `timm`-style implementation fuses QKV, forcing HS-LoRA to be applied to all three projections, which results in **1.5× more parameters** than the PiSSA case. For both setups, HS-LoRA uses per-head rank $r' = 3$, which is non-trivial once multiplied by 12 heads (36 effective heads for CaRA).

| | #param (M) | Cifar100 | Caltech101 | DTD | Flower102 | Pets | SVHN | Sun397 | CameLyon | EuroSAT | Resisc45 | Retinopathy | Clev-Count | Clev-Dist | DMLab | KITTI-Dist | dspr-Loc | dspr-Ori | sNORB-Azim | sNORB-Ele | Group Mean | Overall Mean |
|---|---|---|---|---|---|---|---|---|---|---|---|---|---|---|---|---|---|---|---|---|---|---|
| | | | | Natural | | | | | | Specialized | | | | | | Structured | | | | | | |
| Traditional Fine-Tuning[†] | | | | | | | | | | | | | | | | | | | | | | |
| Linear | - | 63.4 | 85.0 | 63.2 | 97.0 | 86.3 | 36.6 | 51.0 | 78.5 | 87.5 | 68.6 | 74.0 | 34.3 | 30.6 | 33.2 | 55.4 | 12.5 | 20.0 | 9.6 | 19.2 | 57.64 | 52.94 |
| FT | 85.8 | 68.9 | 87.7 | 64.3 | 97.2 | 86.9 | 87.4 | 38.8 | 79.7 | 95.7 | 84.2 | 73.9 | 56.3 | 58.6 | 41.7 | 65.5 | 57.5 | 46.7 | 25.7 | 29.1 | 68.96 | 65.57 |
| PEFT methods[†] | | | | | | | | | | | | | | | | | | | | | | |
| Adapter based[†] | | | | | | | | | | | | | | | | | | | | | | |
| Adapter-256 | 0.27 | 74.1 | 86.1 | 63.2 | 97.7 | 87.0 | 34.6 | 50.8 | 76.3 | 88.0 | 73.1 | 70.5 | 45.7 | 37.4 | 31.2 | 53.2 | 30.3 | 25.4 | 13.8 | 22.1 | 59.95 | 55.82 |
| VPT-Shallow | 0.06 | 77.7 | 86.9 | 62.6 | 97.5 | 87.3 | 74.5 | 51.2 | 78.2 | 92.0 | 75.6 | 72.9 | 50.5 | 58.6 | 40.5 | 67.1 | 68.7 | 36.1 | 20.2 | 34.1 | 67.82 | 64.85 |
| VPT-Deep | 0.53 | **78.8** | 90.8 | 65.8 | 98.0 | 88.3 | 78.1 | 49.6 | 81.8 | 96.1 | 83.4 | 68.4 | 68.5 | 60.0 | 46.5 | 72.8 | 73.6 | 47.9 | 32.9 | 37.8 | 71.96 | 69.43 |
| AdaptFormer | 0.16 | 70.8 | 91.2 | 70.5 | 99.1 | 90.9 | 86.6 | 54.8 | 83.0 | 95.8 | 84.4 | **76.3** | 81.9 | 64.3 | 49.3 | 80.3 | 76.3 | 45.7 | 31.7 | 41.1 | 74.75 | 72.32 |
| SSF | 0.24 | 69.0 | 92.6 | **75.1** | **99.4** | **91.8** | 90.2 | 52.9 | 87.4 | 95.9 | 87.4 | 75.5 | 75.9 | 62.3 | 53.3 | 80.6 | 77.3 | 54.9 | 29.5 | 37.9 | 75.69 | 73.10 |
| RepAdapter | 0.22 | 72.4 | 91.6 | 71.0 | 99.2 | 91.4 | 90.7 | 55.1 | 85.3 | 95.9 | 84.6 | 75.9 | 82.3 | 68.0 | 50.4 | 79.9 | 80.4 | 49.2 | 38.6 | 41.0 | 76.09 | 73.84 |
| NAS based[†] | | | | | | | | | | | | | | | | | | | | | | |
| NOAH | 0.361 | 69.6 | 92.7 | 70.2 | 99.1 | 90.4 | 86.1 | 53.7 | 84.4 | 95.4 | 83.9 | 75.8 | 82.8 | 68.9 | 49.9 | 81.7 | 81.8 | 48.3 | 32.8 | 44.2 | 75.5 | 73.25 |
| LoRA based[†] | | | | | | | | | | | | | | | | | | | | | | |
| LoRA | 0.29 | 67.1 | 91.4 | 69.4 | 98.8 | 90.4 | 85.3 | 54.0 | 84.9 | 95.3 | 84.4 | 73.6 | 82.9 | 69.2 | 49.8 | 78.5 | 75.7 | 47.1 | 31.0 | 44.0 | 74.60 | 72.25 |
| FacT-TT | 0.04 | 71.3 | 89.6 | 70.7 | 98.9 | 91.0 | 87.8 | 54.6 | 85.2 | 95.5 | 83.4 | 75.7 | 82.0 | 69.0 | 49.8 | 80.0 | 79.2 | 48.4 | 34.2 | 41.4 | 75.34 | 73.04 |
| FacT-TK | 0.07 | 70.6 | 90.6 | 70.8 | 99.1 | 90.7 | 88.6 | 54.1 | 84.8 | 96.2 | 84.5 | 75.7 | 82.6 | 68.2 | 49.8 | 80.7 | 80.8 | 47.4 | 33.2 | 43.0 | 75.56 | 73.23 |
| SPT-LoRA | 0.54 | 73.5 | **93.3** | 72.5 | 99.3 | 91.5 | 87.9 | **55.5** | 85.7 | 96.2 | 85.9 | 75.9 | **84.4** | 67.6 | 52.5 | **82.0** | 81.0 | 51.1 | 30.2 | 41.3 | 76.37 | 74.07 |
| CaRA | 0.06 | 71.3 | 91.9 | 71.8 | 99.3 | 91.4 | 90.5 | 54.7 | 86.2 | **96.4** | 86.0 | 75.4 | 83.8 | **69.4** | 51.4 | 81.7 | 80.8 | 47.4 | 35.3 | 44.0 | 76.46 | 74.14 |
| PiSSA (P1) | 0.88 | 67.69 | 91.02 | 71.65 | 98.85 | 90.02 | 89.94 | 54.69 | 85.18 | 95.67 | 86.49 | 74.46 | 83.5 | 62.2 | **53.72** | 78.2 | 79.27 | 44.99 | 33.73 | 28.17 | 74.66 | 72.08 |
| HS+PiSSA (P2) | 0.48 | 67.34 | 91.81 | 71.65 | 98.85 | 91.30 | **91.77** | 54.87 | **88.24** | 96.19 | **87.68** | 76.03 | 83.42 | 62.3 | 53.67 | 80.3 | 79.27 | 47.09 | 34.13 | 29.97 | 75.63 | 72.94 |
| HS+CaRA (P2) | 0.72 | 72.16 | 92.03 | 71.01 | 99.17 | 91.39 | 90.68 | 55.14 | 85.5 | 96.31 | 85.94 | 75.77 | 83.68 | 65.11 | 50.96 | **82.0** | **82.19** | **59.33** | **39.04** | **45.71** | **76.99** | **74.87** |

Table 2: VTAB-1k results on Natural, Specialized, and Structured categories. Rows above `CaRA` are reproduced from the CaRA paper Veeramacheneni et al. (2025) for reference.

**Results.** Table 2 summarises the results. HS-LoRA is not uniformly beneficial: in the **Natural** and **Specialized** categories, improvements are marginal or even slightly negative (PiSSA: +0.53/+1.59 pp; CaRA: +0.09/-0.12 pp). By contrast, in the **Structured** category, HS-LoRA yields consistent gains (PiSSA: +0.80 pp; CaRA: +1.78 pp). For PiSSA, improvements are driven by KITTI-Dist (+2.10), dSprites-Ori (+2.10), and sNORB (Azim +0.40, Ele +1.80), while Clev-Count (-0.08) and DMLab (-0.05) show minor drops. For CaRA, gains come mainly from dSprites-Ori (+11.93), sNORB (Azim +3.74, Ele +1.71), and dSprites-Loc (+1.39), though Clev-Dist drops (-4.29).

**Takeaways.** While the overall mean gains are modest (PiSSA: +0.86 pp; CaRA: +0.73 pp), the improvements concentrate in the Structured category, where several tasks show large boosts despite occasional drops elsewhere. One possible explanation is that ImageNet pretraining already aligns well with Natural and Specialized domains, leaving limited room for head-specific refinements, whereas Structured tasks involve domain shifts that benefit more from disentangling global and head-specific variations.

## 6 Discussion

**Rank efficiency.** Across LoRA-family methods, it is well recognized that simply increasing rank does not yield linear gains—early improvements quickly saturate and, beyond a certain point, addi-

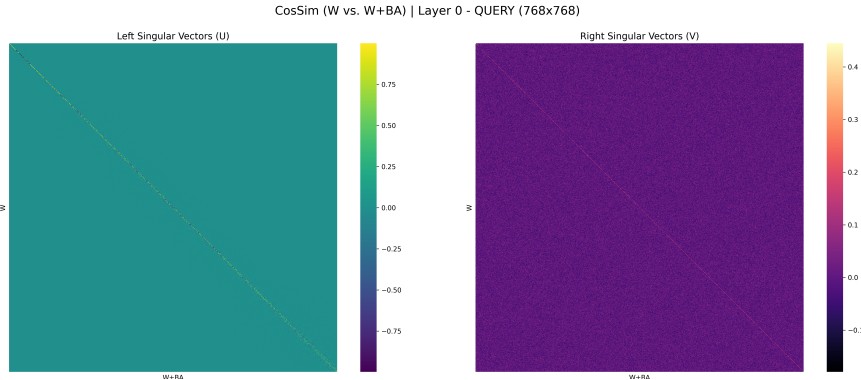

Figure 2: Cosine similarity of singular vectors before vs. after fine-tuning (entire projection matrix). Left: left singular vectors ($U$) remain stable. Right: right singular vectors ($V$) exhibit widespread directional changes with weaker alignment.

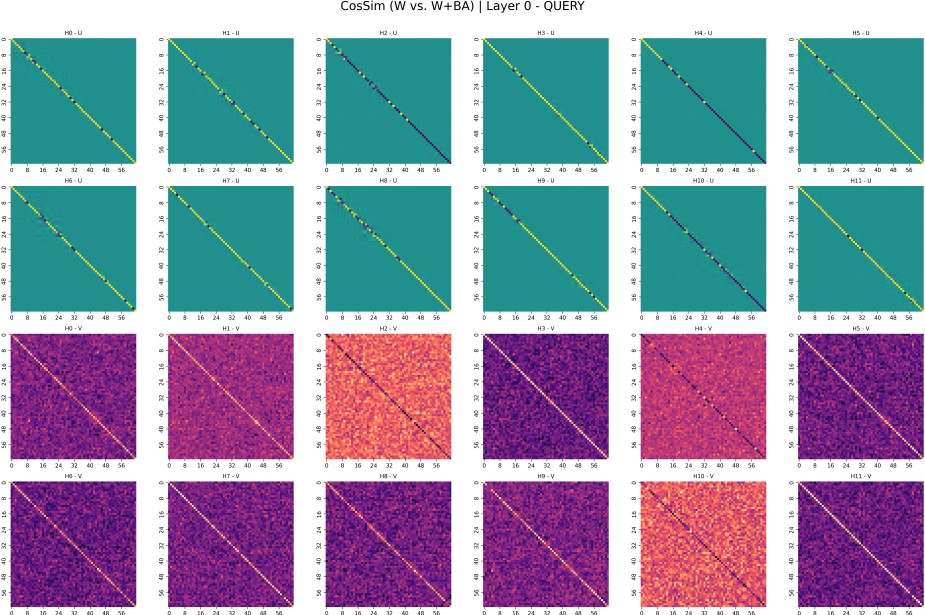

Figure 3: Cosine similarity of singular vectors before vs. after fine-tuning (per-head analysis). Right singular vectors ($V$) show clearer diagonal alignment at the head level, revealing latent structure obscured in the full-matrix view.

tional rank often brings diminishing or even negative returns Hu et al. (2022); Zhang et al. (2023a); Liu et al. (2024). Our findings reinforce this perspective: rather than indiscriminately enlarging rank, Phase 2 HS-LoRA provides a more targeted way to expand the *effective rank* by focusing on head-specific refinements.

**Subspace analysis.** To better understand how LoRA-based fine-tuning affects the internal representations of multi-head attention (MHA) layers, we analyze the change in their weight subspaces before and after fine-tuning. We focus on the query, key, and value projection matrices, which share a similar structural decomposition. For a projection weight $W \in \mathbb{R}^{d \times d}$, we perform singular value decomposition (SVD):

$$W = U\Sigma V^\top,$$

where $U$ and $V$ are the left and right singular vectors, respectively. We then compare the singular vectors before and after fine-tuning using cosine similarity.

*Findings on entire projection matrices.* When analyzing the full projection matrix (e.g., the entire query projection), we observe that the left singular vectors $U$ remain highly aligned before and after fine-tuning, with cosine similarity close to $\pm 1$ along the diagonal (Figure 2, left). This suggests that LoRA updates induce minimal change in the output subspace. In contrast, the right singular vectors $V$ exhibit widespread directional changes, with no clear diagonal alignment (Figure 2, right). Notably, the color scale for the left singular vectors spans nearly the full $[-1, 1]$ range, whereas the right singular vectors are limited to a narrower interval (approximately $[-0.15, 0.43]$). This further emphasizes that the output subspace is extremely stable, while the input subspace undergoes smaller but more chaotic directional changes.

*Findings on per-head sub-blocks.* When the same analysis is performed on each attention head separately (by isolating the head-specific sub-blocks from the projection matrix), a different pattern emerges. The left singular vectors $U$ still remain stable, as in the full-matrix analysis. However, the right singular vectors $V$ now reveal clearer diagonal alignment compared to the noise-like pattern observed at the full-matrix level (Figure 3). This implies that while the aggregate right singular space of the entire projection appears chaotic, each head preserves more structured directions individually.

*Implications.* These findings suggest that LoRA-based updates primarily modify the input subspace (right singular vectors) rather than the output subspace (left singular vectors). Moreover, because the right singular space becomes highly entangled when analyzed at the full-matrix level, significant parameter capacity is required to adapt these directions effectively. The head-level analysis shows that the underlying structure of the right singular vectors is more coherent at finer granularity, which implies that targeting these subspaces more explicitly could lead to more efficient parameter usage. This directly motivates Phase 2 HS-LoRA: by operating at the head level, it exploits structured subspaces that full-matrix adaptation obscures.

**Synergy with global methods.** PiSSA exemplifies this complementarity. Its SVD-based initialization prioritizes dominant global subspaces, while Phase 2 HS-LoRA injects head-specific residual variations that Phase 1 alone cannot capture. This pairing achieves the largest improvements in the Specialized category (+2.09 pp), underscoring that methods designed for global adaptation are natural companions for our two-phase design.

**Domain-specific effects.** Performance gains are not uniform across domains. Natural tasks, being closest to ImageNet pretraining, leave little room for improvement. Structured tasks, by contrast, show the most pronounced benefits (e.g., CaRA +7.59 pp), while Specialized tasks see moderate gains. We interpret this as evidence that HS-LoRA is especially effective under significant domain shifts, where individual heads must reorient to capture new geometric or relational patterns. In such cases, Phase 2 acts as an extension of usable rank beyond the plateau observed in Phase 1.

**Practical implications.** Although HS-LoRA can increase parameter counts, it remains far more efficient than full fine-tuning. When parameters must be kept low, Phase 1 global adaptation may suffice; but when additional head-specific flexibility is warranted, Phase 2 offers a compelling and selective path forward. Rather than a universal booster, HS-LoRA should be seen as a domain-sensitive strategy: most impactful when distribution shifts are large, and less necessary when target tasks remain close to the pretraining domain.

## 7 CONCLUSION

In this work, we introduced **Two-Phase Head-Specific LoRA (HS-LoRA)**, which disentangles adaptation into global and head-specific subspaces. Experiments on VTAB-1k demonstrate that HS-LoRA yields consistent gains over strong Phase 1 baselines, particularly under domain shifts where head individuality is crucial. The method proves especially effective in Structured and Specialized domains, complementing global (e.g., PiSSA) and tensorized (e.g., CaRA) approaches. Importantly, HS-LoRA retains the deployment efficiency of the LoRA family: adapters can be merged into base weights with no inference overhead. Overall, HS-LoRA reveals a new axis of parameter-efficient fine-tuning, leveraging head individuality. We hope it inspires future partition-aware PEFT design.

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

## A    APPENDIX

**The use of LLMs.** We use LLMs only for minor language editing, including adjustments to word choices and clarity. LLMs played no role in the research design, analysis, interpretation, or manuscript preparation.

