# OpenReview forum: "Two-Phase Head-Specific LoRA: Balancing Global and Local Adaptation in Multi-Head Attention"
_ICLR.cc/2026/Conference — ICLR 2026 Conference Withdrawn Submission_

### Official Review · Reviewer_8i1U · 2025-10-22

**Soundness:** 1
**Presentation:** 3
**Contribution:** 2
**Rating:** 4
**Confidence:** 3

**Summary:**

This work is motivated by the observation that different attention heads serve distinct functions, and it proposes a Two-Phase Head-Specific LoRA method. Specifically, the method first fine-tunes the model using LoRA (or its variants) in a global manner, and then further refines the updated model with head-specific LoRA.

**Strengths:**

1. The method is straightforward and easy to implement.
2. The manuscript is well-written and easy to understand.

**Weaknesses:**

1. The experiments lack details on certain hyperparameters, such as the number of training epochs and learning rates.
2. Compared to the CaRA method, the proposed method introduces many more trainable parameters, yet the performance improvement is very limited, raising concerns about the effectiveness of the proposed approach.
3. The proposed two-phase method lacks intuitive or theoretical motivation, making it difficult to determine whether the observed improvement stems from the increased number of trainable parameters or from the method itself.

**Questions:**

1. To my knowledge, some prior work [1, 2] has explored fine-tuning only a subset of attention heads, but such comparisons are not presented in this paper.
[1] HeadMap: Locating and Enhancing Knowledge Circuits in LLMs, ICLR 2025
[2] Interpreting and Improving Large Language Models in Arithmetic Calculation, ICML 2024.

---

### Official Review · Reviewer_G2sk · 2025-10-27

**Soundness:** 3
**Presentation:** 3
**Contribution:** 3
**Rating:** 2
**Confidence:** 4

**Summary:**

The authors propose Two-Phase Head-Specific LoRA (HS-LoRA), a method that decomposes adaptation into two phases. In the first phase, a global adapter, implemented through any standard LoRA-based PEFT method such as LoRA, PiSSA, DoRA, or AdaLoRA, captures broad domain-shift information common across all attention heads. In the second phase, head-specific adapters are introduced to refine each head’s projections, recovering fine-grained adaptation suppressed by the global phase. The approach shows moderate but consistent improvements on the VTAB-1k benchmark and demonstrates complementary effects when combined with existing LoRA variants such as PiSSA and CaRA. The design maintains simplicity, compatibility with existing methods, and no inference-time overhead.

**Strengths:**

Simplicity of the intuition and method: the two-phase design is conceptually simple and easy to understand, following a clear logic of global-to-local refinement.

Implementation practicality: the method is compatible with any LoRA-family approach and introduces negligible additional overhead, preserving the efficiency and mergeability properties of standard LoRA.

Interesting subspace analysis: the authors include an insightful SVD-based study that supports their intuition: at the head level, right singular spaces show more coherent structure than when analyzed globally.

**Weaknesses:**

Limited and inconsistent performance gains: while the head-specific phase occasionally boosts performance, particularly for structured tasks, the overall mean improvements are modest (+0.7–0.8 pp). The effect is not uniform across datasets, with occasional drops undermining the generality of the method. Although the authors discuss domain-shift sensitivity as a potential explanation, the observed gains remain limited.

Lack of strong ablations and empirical support for claims:
  * Lines 85–86: They claim LoRA performs worse on fused QKV projections but provide no quantitative evidence.
  * Line 318: The chosen per-head rank r' is not justified or compared to alternatives.
  * Lines 331–332: They apply HS-LoRA only to query and value projections, citing “minimal contribution” from keys, yet no citation or ablation supports this.
  * Line 420: They state that increasing per-head rank is not beneficial, again without supporting experiments.

Limited experimental scope: experiments focus solely on ViT-B/16 in the vision domain (VTAB-1k). No evidence is given for applicability to text or multimodal models, leaving the generalization claim untested.

No statistical analysis or significance tests: the results are presented as single averages without standard deviations or multiple runs, making it unclear whether improvements are statistically meaningful.

Uniform LoRA type across heads: the method applies identical LoRA configurations to all heads. Different heads may capture distinct types of information (local vs global, spatial vs semantic), and using heterogeneous LoRA variants or rank allocations could potentially yield stronger results. The authors do not explore this aspect, which limits the depth of the “head-specific” idea.

No direct comparison with single-phase head-specific LoRA: the method extends a prior head-specific LoRA idea but does not present explicit experiments isolating the benefit of adding the global phase.

**Questions:**

1. How sensitive is HS-LoRA to the per-head rank (r′)?
2. Have you considered using different LoRA variants per head or adaptive rank selection across heads?
3. How does the method compare with a single-phase head-specific LoRA trained directly (without a global phase)?

---

### Official Review · Reviewer_FcQi · 2025-10-29

**Soundness:** 2
**Presentation:** 2
**Contribution:** 2
**Rating:** 2
**Confidence:** 4

**Summary:**

This paper introduces Two-Phase Head-Specific LoRA (HS-LoRA), a novel parameter-efficient fine-tuning (PEFT) method that decomposes adaptation into two stages: (1) a global low-rank update applied across the entire projection matrix to capture broad, domain-shared shifts, and (2) head-specific low-rank refinements applied per attention head to restore individuality suppressed by the global update.

The motivation arises from the observation that conventional LoRA assumes uniform adaptation across all attention heads, ignoring their functional diversity. The authors show through singular value decomposition (SVD) analysis that LoRA primarily perturbs the *input subspace* (right singular vectors), and that this perturbation appears chaotic when viewed at the full-matrix level but reveals structured, coherent variations when examined per head.

Experiments on the VTAB-1k benchmark shows that HS-LoRA improves over global LoRA variants (LoRA, AdaLoRA, DoRA, PiSSA, CaRA), especially on Structured tasks (e.g., dSprites, sNORB, KITTI-Dist).

**Strengths:**

1. **Simplicity and Clarity of Presentation.**

    The proposed method is conceptually simple and well-articulated. The description of the two-phase design is clear and easy to follow, making the overall approach accessible to readers.

2. **Interpretability Contribution**

    The cosine-similarity visualizations of singular vectors (Figures 2–3) give rare, interpretable insights into how LoRA affects internal representations of attention layers.

**Weaknesses:**

1. **Limited performance improvement.**

    The reported improvements are concentrated primarily in Structured tasks, while the gains on Natural and Specialized tasks are minor or even negative. Although the authors provide some qualitative explanations, the overall performance enhancement remains marginal. Moreover, HS-LoRA *reduces* performance in certain cases, for instance, CaRA drops by 4.39% on Clev-Dist.

2. **Lack of implementation details.**

    Key training details are missing, including the specific training steps for the first and second phases. Additionally, the model configurations corresponding to Table 2 are not described.

3. **Hyperparameter Sensitivity**

    The effectiveness of HS-LoRA may depends on the selection of the global rank and per-head rank. However, the paper does not include an ablation study or sensitivity analysis to examine how these hyperparameters influence performance.

4. **Some Typos:** line 318, Table 5.2 should be Table 1.
5. **Overclaim:** In Table 2, the reported parameter count for HS-PiSSA (0.48M) appears to include only the Phase-2 parameters. Since Phase 2 is applied based on Phase 1, even the phase 1 LoRA can be merged into the weight, the total training parameter count should reflect the combined contributions of both phases.

**Questions:**

1. Why the result of SVHN are inconsistent in Table 1 (90.39) and Table 2 (91.77)?
2. How is the training dynamics of HS-LoRA? Are different training steps of phase 1 and phase 2 highly influence the performance?
3. Since Phase 2 are continual training after Phase 1, is the comparsion between HS-PiSSA and PiSSA fair? They have different overall training step. Could you conduct experiments that training under the same overall training steps?
4. How stable is training when the number of heads increases (e.g., ViT-L/32 or large LLMs)?

    Does Phase-2 introduce optimization difficulty or gradient imbalance?

5. Could the authors provide a quantitative measure of “head individuality” or subspace overlap before and after applying HS-LoRA?

---

### Official Review · Reviewer_Wic3 · 2025-11-01

**Soundness:** 3
**Presentation:** 3
**Contribution:** 3
**Rating:** 4
**Confidence:** 4

**Summary:**

The paper introduces Two-Phase Head-Specific LoRA (HS-LoRA), a novel framework aimed at improving Low-Rank Adaptation (LoRA), which is widely used for parameter-efficient fine-tuning of pre-trained models. LoRA, while effective, applies a single low-rank update across all attention heads in multi-head attention models, which may overlook the diverse roles of individual heads. The paper addresses this limitation by proposing a two-phase process.

In Phase 1, a global low-rank update is applied across all attention heads, capturing domain-shift information that is common across heads. This phase is efficient but may not fully capture the individuality of each attention head. In Phase 2, lightweight head-specific adapters are applied to refine residual variations and recover the individuality of each attention head that was suppressed by the global update. This approach strikes a balance between global efficiency and head-specific expressiveness.

**Strengths:**

The two-phase method optimizes the adaptation of large models by combining global and head-specific updates, thus offering better performance without excessive parameter growth.

The method excels particularly in handling domain shifts in tasks, where individual attention heads need to adapt to different patterns (e.g., in structured tasks).

When combined with other global adaptation methods like PiSSA, HS-LoRA provides complementary improvements, enhancing the effectiveness of LoRA-style methods.

Like standard LoRA, HS-LoRA allows the low-rank adapters to be merged back into the original weights during inference, avoiding additional computational overhead.

**Weaknesses:**

While the method is efficient, Phase 2's per-head updates can increase the parameter count, especially when there are many attention heads.

In certain domains, particularly tasks close to pretraining domains (like natural tasks), the benefits of the head-specific updates are less pronounced.

When using fully independent adapters for each head in Phase 2, some redundancy is introduced, as many adaptation signals are shared across heads.

**Questions:**

Can you provide a Pareto curve analysis showing the trade-off between the increase in parameters (due to per-head LoRA) and performance improvements across various tasks?

How does HS-LoRA perform when applied to NLP tasks such as SuperGLUE, MATH, or LongBench? What challenges might arise in these domains?

How would HS-LoRA adapt to vision-generation or diffusion models? Are any modifications needed to support these tasks?

How does HS-LoRA compare to DyLoRA and AdaLoRA in terms of flexibility and adaptation to different domain shifts?

How does HS-LoRA compare with PiLoRA in balancing parameter efficiency with capturing head-specific dynamics?

Can you provide a comparison table for training time, memory usage, and energy consumption of HS-LoRA against LoRA, DyLoRA, AdaLoRA, and PiLoRA?

---

### Note · Authors · 2025-11-24

I have read and agree with the venue's withdrawal policy on behalf of myself and my co-authors.